# Transcriptome Analysis Provides Insights into the Mechanism of the Transcription Factor *AaCrz1* Regulating the Infection Structure Formation of *Alternaria alternata* Induced by Pear Peel Wax Signal

**DOI:** 10.3390/ijms252211950

**Published:** 2024-11-07

**Authors:** Yangyang Yang, Pengdong Xie, Yuanping Nan, Xiaobin Xu, Jing Yuan, Yongcai Li, Yang Bi, Dov Prusky

**Affiliations:** 1College of Food Science and Engineering, Gansu Agricultural University, Lanzhou 730070, China; yanggsau2021@163.com (Y.Y.); xpd_19961008@163.com (P.X.); 17899316118@163.com (Y.N.); xiaobin04182023@163.com (X.X.); yj15214029875@163.com (J.Y.); biyang@gsau.com (Y.B.); 2Department of Postharvest and Food Science, Agricultural Research Organization, Volcani Center, Rishon LeZion 7505101, Israel; dovprusk@volcani.agri.gov

**Keywords:** transcriptome sequencing, *Alternaria alternata*, transcription factor *AaCrz1*, infection structure formation, regulatory mechanisms

## Abstract

*Alternaria alternata*, a causal agent of pear black spot, can recognize and respond to physicochemical signals from fruit surfaces through an intricate signaling network to initiate infection. *Crz1* is an important transcription factor downstream of the calcium signaling pathway. In this study, we first investigated the infection structure formation process of the wild type (WT) and *ΔAaCrz1* strains induced by the cuticular wax of the “Zaosu” pear by microscopic observation. We found that the infection process was delayed and the rate of appressorium formation and infection hyphae formation was significantly decreased in the *ΔAaCrz1* strain. RNA-seq of WT and *ΔAaCrz1* strains was analyzed after 6 h of induction with pear wax. A total of 893 up-regulated and 534 down-regulated genes were identified. Among them, genes related to cell wall degrading enzymes, ABC transporters, and ion homeostasis were down-regulated, and the autophagy pathway was induced and activated. In addition, disruption to the intracellular antioxidant system was also found after *AaCrz1* knockdown. In summary, this study provides new information on the mechanism of the transcription factor *AaCrz1* in the regulation of infection structure formation of *A. alternata* induced by pear peel wax signal, which can be used to develop new strategies for controlling fungal diseases in the future.

## 1. Introduction

*Alternaria alternata*, an important postharvest pathogen with a wide range of hosts, can cause black spot disease in various fruits and vegetables such as pears, melons, citrus, peaches, and potatoes [1,2]. *A. alternata*, as a typical latent infection fungus, initiates infection when it senses favorable signals, starting with spore germination to form germ tubes that subsequently differentiate to appressoria [3], which can invade the fruit through pressure-mediated penetration of the intact host cuticle and outer epidermal cell walls or lenticel and epidermal wounds, employing various strategies that have evolved to invade fruit effectively [4]. Due to the complexity and diversity of the infection pathways, infection modes, pathogenic factors, and regulatory mechanisms of *A. alternata*, it is of great significance to comprehensively reveal the molecular pathogenic mechanisms of *A. alternata* in order to control postharvest black spots in fruit.

Plant surface cuticular wax is a complex mixture of hydrophobic substances, whose components and their degradation products can be utilized by phytopathogenic fungi as part of the signals that trigger spore germination and infection structure differentiation [5,6]. Intracellular signal transduction pathways are involved in the process of pathogenic fungi recognition and perception of external physical and chemical stimuli to initiate infection. These signal transduction pathways, including the cyclic adenosine monophosphate-protein kinase A (cAMP-PKA) pathway, mitogen-activated protein kinase (MAPK) pathway, and calcium signaling pathway, can induce cellular responses through membrane-to-membrane transduction and intracellular phosphorylation cascades [7,8,9,10,11,12]. In particular, changes in intracellular Ca^2+^ concentration have been found to be an important regulatory signal for a variety of cellular metabolic activities [13]. Current research reports on the calcium signaling pathway in fungal species have mainly focused on pathway proteins such as phospholipase C, calmodulin, and calcineurin [8,14]. In addition, *Crz1* is an important zinc finger transcription factor downstream of the calcium signaling pathway in pathogenic fungi. The Cys2His2 zinc finger structural domain of *Crz1* binds to the promoter elements of target genes and activates target gene expression [15,16]. The function of *Crz1* homologues has been reported in many fungi. In *Saccharomyces cerevisiae*, *Crz1* regulated tolerance to high concentrations of Ca^2+^ by regulating genes such as ATPase *PMC1* [17]. *Crz1* positively regulates the virulence of human and plant pathogenic fungi. The *Crz1* deficiency reduced the virulence of *Magnaporthe grisea* and *Botrytis cinerea* by affecting appressorium or sclerotium formation [18,19]. The virulence of *Candida glabrata*, *Aspergillus fumigatus*, and *Cryptococcus neoformans* [20,21,22] decreased as a result of the deletion of *Crz1*. It has also been observed that knocking out *Crz1* results in impaired fungal growth, reduced conidia [23], inhibition of toxin synthesis [23], and impaired cell wall integrity [10]. The function and possible regulatory mechanism of *Crz1* has always been a hot topic in the field of fungal research.

In this study, we observed the infection structure differentiation and penetration process of wild type (WT) and *ΔAaCrz1* on onion epidermis coated with pear peel wax. We found that the infection structure of *ΔAaCrz1* was damaged during the early infection stage. However, the specific regulatory mechanism of how *AaCrz1* participates in the structural differentiation and pathogenicity of *A. alternata* by responding to cuticular wax signals remains unclear. Thus, RNA-seq was used for analysis of the differentially expressed genes (DEGs) and differential metabolic pathways of *ΔAaCrz1* compared to WT after 6 h of induction with the cuticular wax of a “Zaosu” pear. The data showed that genes related to cell wall degrading enzymes, ABC transporters, and ion homeostasis were down-regulated, and the autophagy pathway was induced and activated. In addition, the intracellular antioxidant system was also found to be destroyed after *AaCrz1* knockout. These results greatly enriched our understanding of the regulatory mechanism of *AaCrz1* on *A. alternata* infection structure formation induced by pear peel wax signal, and this RNA-seq result provides data support for the search for potential genes or proteins in *A. alternata* as targets for black spot disease control.

## 2. Results

### 2.1. AaCrz1 Deletion Affects the Infection Structure Differentiation of A. alternata on the Onion Epidermis Coated with Fruit Wax Extract

To observe the complete infection structure differentiation and penetration process of *A. alternata*, the spore suspension of the WT, *ΔAaCrz1*, and *ΔAaCrz1-C* were dropped on onion skin coated with fruit wax for observation. The recombinant pC-NEO vector containing the *AaCrz1* gene sequence was transferred into *ΔAaCrz1* by PEG-mediated protoplast transformation to obtain the complementary strain *ΔAaCrz1-C*. The results showed that *A. alternata* conidia were rod-shaped, ovoid, or ellipsoid with longitudinal and transverse septa (Figure 1A). The deletion of *AaCrz1* reduced the differentiation ability of the infection structure in *A. alternata*. Compared to WT and *ΔAaCrz1-C*, the *ΔAaCrz1* spore germination was decreased by 54.05% and 52.77%, respectively, at 4 h of incubation (Figure 1B). The appressorium formation and infection hyphae formation rate of *ΔAaCrz1* were decreased by 51.34% and 39.98%, respectively, at 8 h of incubation (Figure 1C,D). However, *ΔAaCrz1-C* restored the defective phenotype of the *ΔAaCrz1* mutant, and the appressorium formation and infection hyphae formation were significantly improved. Moreover, there was no significant difference between the *ΔAaCrz1-C* and WT strains during this process.

### 2.2. Gene Expression Abundance Distribution and Sample Relationship Analysis

In order to explore the specific regulatory mechanism of AaCrz1 on the structural differentiation and pathogenicity of *A. alternata* upon responding to cuticular wax signals, RNA-seq was used to analyze the differentially expressed genes (DEGs) and differential metabolic pathways of *ΔAaCrz1* compared with the WT 6 h after incubating with cuticular wax from the “Zaosu” pear. The FPKM density distribution comparison diagram of six samples of WT and *ΔAaCrz1* strains showed that the gene expression level spanned six orders of magnitude from 10^−2^ to 10^4^ (Figure 2A). The results of the violin plots showed that the expression levels of the six samples were consistent (Figure 2B). According to the principal component analysis (PCA), a tight clustering of samples in the same group indicated a good level of repeatability of the samples in the same group. Notable dispersion between samples from different treatments indicated that the samples correctly reflected the biological differences between different groups (Figure 2C). The results of sample correlation analysis showed that the duplicate samples within the same group showed a highly significant positive correlation, and the correlation coefficients were in the range of 0.782 to 0.994, indicating that the reproducibility between the samples was good and the quality of sequencing was reliable (Figure 2D).

### 2.3. DEGs Analysis

To identify genes that exhibit significant changes in expression levels between the wild-type strain (WT) and *ΔAaCrz1* (M) groups were screened with FDR < 0.05 and |log_2_FC| > 1. The results showed that compared with WT, *ΔAaCrz1* had 1427 differentially expressed genes (DEGs), including 534 (37.42%) down-regulated genes and 893 (62.58%) up-regulated genes, indicating that the deletion of *AaCrz1* gene had a great effect on the gene expression level of *A. alternata* under pear fruit wax induction (Figure 3A). Specifically, the overall distribution of these genes is shown in the volcano plot, with yellow dots representing down-regulated DEGs and blue dots representing up-regulated DEGs (Figure 3B).

### 2.4. GO Enrichment Analysis

DEGs data were functionally categorized using Gene Ontology, which contains three main categories: biological process, molecular function, and cellular composition (Figure 4A). Since some DEGs can be categorized into different category branches, the total number of DEGs in the GO classification can be larger than the actual total number of DEGs. In this study, GO enrichment analysis found a total of 19 biological process terms, involving 1006 up-regulated DEGs and 723 down-regulated DEGs; seven molecular functional terms, involving 458 up-regulated DEGs and 332 down-regulated DEGs; and 14 cell component terms, involving 438 up-regulated DEGs and 430 down-regulated DEGs. The top 20 most significantly enriched GO terms in −log_10_(*Q*-value) are shown in Figure 4B. The top three GO terms in cell components were all plasma membrane related, including the intrinsic component of membranes, membrane parts, and membranes as a whole. Two of the three terms of biological processes were enriched and annotated as metabolic processes: the phenol-containing compound metabolic process, and the melanin metabolic process. DEGs were also significantly enriched in drug transport. Among the 14 terms of molecular function, 5 terms related to enzyme activity were significantly enriched, with the most significant differences in oxidoreductase activity and potassium-transporting ATPase activity. Five terms related to transporters were significantly enriched, with the most significant differences in potassium ion transmembrane transporter activity and metal ion transmembrane transporter activity. In addition, four terms related to binding proteins were enriched.

### 2.5. KEGG Pathway Enrichment Analysis

The KEGG pathway classification of DEGs was mainly enriched in five categories: metabolism, genetic information processing, cellular processes, environmental information processing, and organismal systems (Figure 5A). In metabolism, DEGs were enriched in 11 KEGG pathway annotations, including primary metabolism (carbohydrate, amino acid, lipid, energy, and nucleotide metabolism), secondary metabolite metabolism (cofactor and vitamin metabolism, terpene, and polyketide metabolism), glycan biosynthesis and metabolism, and biosynthesis of other secondary metabolites. In genetic information processing, DEGs were enriched for four annotations, namely folding, sorting, and degradation, replication and repair, translation, and transcription. Two annotations of transport and decomposition as well as cell growth and death were enriched during cellular processes. Two annotations of signal transduction and membrane transport were enriched in the process of environmental information. Only the annotation of aging was enriched in biological systems.

Subsequently, five annotations of the green underscore in Figure 5A were selected: carbohydrate metabolism, folding, sorting, and degradation, transport and catabolism, membrane transport, and aging. The five pathways most significantly enriched in −log_10_(*Q*-value) under these annotations are shown in Figure 5B, with different colors representing different categories. In carbohydrate metabolism, starch and sucrose metabolism was significantly enriched. Ubiquitin-mediated proteolysis was the most important enrichment pathway during folding, sorting, and degradation. In transport and catabolism, the most important pathway was autophagy (others). In terms of membrane transport, ABC transporters was the most significantly enriched pathway. In terms of aging, the longevity regulatory pathway (multiple species) was the most significantly enriched pathway.

### 2.6. Characterization of DEGs in A. alternata at Early Stages of Infection

Starch and sucrose metabolism, autophagy-other, and ABC transporter proteins, which relate to fungi pathogenicity, were selected for analysis (Figure 5B). The DEGs involved in these pathways are summarized in Appendix A. In addition, information on the DEGs involved in regulation by *AaCrz1* is presented in Appendix A.

#### 2.6.1. Analysis of DEGs Involved in the Selected Pathway

Genes related to starch and sucrose metabolism were downregulated in *ΔAaCrz1* compared to WT (Appendix A). They contained genes belonging to cell wall degrading enzymes (CWDEs), including *EXG1* (ncbi_29108996), *bglF* (ncbi_29120998), and *celb* (ncbi_29116806), which encode the synthesis of 1,3-*β*-glucosidase, *β*-glucosidase, and endoglucanase, respectively. Additionally, two genes down-regulated in this pathway were identified as being involved in trehalose metabolism, including *treh* (ncbi_29119646) and *OTS1* (ncbi_29109130), encoding trehalose 6-phosphate synthase and *α*-trehalase, respectively (Figure 6). DEGs were up-regulated in autophagy-others, including *ATG4* (ncbi_29109885), *ATG9* (ncbi_29111274), and *ATG1* (ncbi_29117977). Among the ABC transporters, *atrA* (ncbi_29117678), and *atrB* (ncbi_29110931), which belong to the ABCG subfamily, were significantly down-regulated.

#### 2.6.2. Analysis of DEGs Involved in the Regulation of AaCrz1

According to the gene function of GO annotation, DEGs possibly regulated by *AaCrz1* were analyzed. Among the cell wall degrading enzymes, in addition to the three genes involved in starch and sucrose metabolism listed in Appendix A, the down-regulated DEGs included a cutinase gene *cutA* (ncbi_29117201), and four endoglucanase genes (ncbi_29116806, ncbi_29120322, ncbi_29115410, and ncbi_29111957). Pathogenic fungi have evolved a complete antioxidant system to maintain ROS balance in vivo. RNA-seq results showed that *cat-1* (ncbi_29109411), *catB* (ncbi_29114347), and *CAT1* (ncbi_29121404) genes were significantly down-regulated in *ΔAaCrz1*, and the *sod-2* (ncbi_29111723) gene was significantly up-regulated. In addition, *Erg3* (ncbi_29116487) was found to be significantly down-regulated in the transcriptome, which would result in the alteration of cell membrane permeability, allowing exogenous H_2_O_2_ to enter the cell. Thus, the transcriptome results showed that *AaCrz1* deletion resulted in excessive accumulation of intracellular ROS. Ion homeostasis is essential for maintaining normal physiological metabolism in pathogenic fungi. RNA-seq results showed that the expression of *pmc1* (ncbi_29117801, ncbi_29120094), *VCX1* (ncbi_29118340, ncbi_29119593), and *RCAN2* (ncbi_29109809) were significantly down-regulated in *ΔAaCrz1* compared with WT (Appendix A).

### 2.7. Identification of Proteins Interacting with AaCrz1

Using the STRING protein database, the differentially expressed proteins interacting with AaCrz1 were predicted, and an interaction network diagram was constructed using Cytoscape. As shown in Figure 7, the top 30 interacting proteins with the highest score were screened out. The results showed that of the hexose transporter, inorganic phosphate transporter, etc., six proteins (yellow) among the proteins interacting with AaCrz1 were encoded by upregulated genes, which accounted for 20% of the total interacting proteins. The expression of 24 proteins (blue), including calcium homeostasis regulating proteins, antioxidant enzymes, drug transporters, etc., was encoded by downregulated genes, which accounted for 80% of the total interacting proteins. The motif sequence information of the transcription factor CRZ1 in *S. cerevisiae* was used to predict the binding sites between these 30 proteins interacting with AaCrz1, and the results are shown in Appendix A.

### 2.8. Validation of Intracellular ROS Accumulation

Elevated levels of ROS can damage fungal cell membranes and cause electrolyte leakage, ultimately leading to cell death. Therefore, fungi generate various enzymes, including superoxide dismutase (SOD) and catalase (CAT), to combat excessive ROS and counter host defense mechanisms. Fluorescence staining showed that the green fluorescence emitted by the WT spores was sparse and dim, while that of the *ΔAaCrz1* spores was dense and bright (Figure 8A). The fluorescence intensity of the *ΔAaCrz1* spores was significantly higher than that of the WT (Figure 8B). Compared to the WT, the *ΔAaCrz1* significantly increased the H_2_O_2_ content by 1.9-fold (Figure 8C). In addition, the expression of *AaSOD* (ncbi_29111723), which is responsible for the disproportionation of O_2_^•−^ to more stable H_2_O_2_, in *ΔAaCrz1* was significantly higher than that in WT. However, the expression of *AaCAT* (ncbi_29121404), an important H_2_O_2_ scavenging enzyme, was significantly decreased. In addition, the expression of the *AaErg3* (ncbi_29116487) gene decreased after *AaCrz1* knockout (Figure 8D), which further promoted intracellular H_2_O_2_ content.

### 2.9. Validation of Gene Expression Associated with Cell Wall Degrading Enzymes, ABC Transporters, Autophagy, and Calcium Homeostasis

In addition to the above ROS-related genes, 11 DEGs were selected for qRT-PCR validation in order to verify the reliability of the RNA-seq data. Among them, four cell wall degrading enzyme genes, *Aacelb* (ncbi_29116806), *AaccutA* (ncbi_29117201), *AabglF* (ncbi_29120998), and *AaEXG1* (ncbi_29108996), were significantly down-regulated in *ΔAaCrz1* (Figure 9A). The expression of two genes belonging to the ABCG transporter family, *AaatrA* (ncbi_29117678) and *AaatrB* (ncbi_29110931), was inhibited (Figure 9B). The autophagy genes *AaATG4* (ncbi_29109885) and *AaATG9* (ncbi_29111274) were significantly up-regulated in *ΔAaCrz1* (Figure 9C). The expression of genes *AaVCX1* (ncbi_29118340), *Aapmc1* (ncbi_29117801), and *AaRCAN2* (ncbi_29109809), which are responsible for maintaining intracellular calcium homeostasis, was significantly inhibited (Figure 9D).

The relative expression of the 14 DEGs determined by RT-qPCR showed a good linear relationship with their expression in the transcriptome with a correlation coefficient of 0.9425, indicating that the transcriptome data were reliable and accurate (Figure 10).

## 3. Discussion

*A. alternata* is currently recognized as a cosmopolitan pathogenic fungus with a broad host range, causing disease in fruit and vegetables such as pears, citrus, broccoli, carrots, and tobacco, as well as many ornamental and weedy plants [1,24,25]. As a latent fungus, the infection structure differentiation of *A. alternata* plays a key role in the disease cycle [3]. Fruit surface wax plays an important role in this process [26,27,28,29]. Previous studies have shown that the highly hydrophobic surface of wax can induce fungi to produce appressoria [3,8,9,10,12,14,27,30]. The components of wax have also proven to be beneficial to the differentiation of fungal infection structures [26,27]. C26 aldehyde and C28 aldehyde in barley and wheat leaf wax are the most effective components in promoting the development of *Blumeria graminis* infections [26,28,29,31], indicating complex morphological structures and compositions of plant surface wax, as physical and chemical cues regulate various physiological and biochemical processes during pathogenic fungi infection.

The calcineurin responsive transcription factor *Crz1* is involved in the regulation of the growth and pathogenicity of a variety of plant pathogenic fungi [19,32,33,34,35]. Our previous study found that *AaCrz1*, an important transcription factor in the calcium signaling pathway, was involved in the regulation of growth and development, calcium homeostasis, cell wall integrity, and pathogenicity in *A. alternata* [10,11]. In this experiment, we observed the effect of *AaCrz1* knockout on the infection structure formation in *A. alternata*. The results showed that compared with WT, the *ΔAaCrz1* spore germination was delayed and the rate of appressorium formation and infection hyphae formation was significantly decreased on the wax-coated onion epidermis, which directly demonstrated that the transcription factor *AaCrz1* played a positive regulatory role in response to wax-induced *A. alternata* infection structure formation. This result was also consistent with previous results in *B. graminis*, *Magnaporthe grisea*, and *Puccinia graminis* [31,36,37,38]. However, there were no differences in the rates of spore germination and appressorium formation in *ΔMoCRZ1* [20]. Therefore, the regulatory effect of *Crz1* on infection structure formation varies with different fungal species. The specific regulatory mechanism of *AaCrz1* on *A. alternata* infection structure formation needs further investigation.

To reveal the physiological function of *AaCrz1* in the process of wax-induced infection structure formation in *A. alternata*, RNA-seq with a reference genome was performed on WT and *ΔAaCrz1* after pear wax induction for 6 h. A total of 1427 differentially expressed genes were screened, indicating that the *AaCrz1* deletion had a great effect on *A. alternata* gene expression (Figure 3A). GO enrichment analysis of DEGs can annotate the classification of gene GO function and analyze the significance of GO function enrichment. GO enrichment analysis showed that DEGs differed most significantly in three aspects (Figure 4B). The analysis results based on KEGG are helpful for further analysis of some significant enrichment pathway annotations in DEGs. The five most significantly enriched pathways are shown in Figure 5B.

The key to the successful infection of plant pathogenic fungi is to break through the obstacles of the host cell wall. Therefore, fungi need to secrete a variety of cell wall degrading enzymes to destroy various polysaccharides in the host cell wall. The role of CWDE as a pathogenic factor in pathogenic fungi has been verified [39,40,41]. RNA-seq data showed that the expression levels of cell wall degrading enzymes were significantly down-regulated (Figure 6). Our experiment provided reference information on the cell wall-degrading enzymes (CWDEs) genes, which will help to identify the pathogenic factors in *A. alternata*. ABC transporters are membrane-integrated proteins that catalyze the hydrolysis of ATP and utilize the energy generated by the hydrolysis to facilitate the transmembrane transport of substrates [42]. It is well known that ABCG plays an important role in the multidrug resistance of fungi [43,44,45]. The expression of two ABC transporter genes was significantly down-regulated in *ΔAaCrz1*.

The production of ROS as a means of resisting pathogens is an early defence response of plants to pathogenic fungal infection [46]. In response, pathogenic fungi have also formed a powerful antioxidant system, including superoxide dismutase (SOD), catalase (CAT), peroxidase (POD), glutathione (GLU), and thioredoxin [47,48]. Peroxisomes can maintain the dynamic balance of ROS in cells by producing and removing ROS. When *Fusarium graminearum* is unable to maintain intracellular ROS homeostasis due to dysfunctional peroxisomes, the level of ROS accumulation in the mycelium is elevated and it is more sensitive to reactive oxygen bursts produced by the plant during the infection process [32]. In this study, *ΔAaCrz1* showed higher ROS fluorescence intensity and H_2_O_2_ content, suggesting that *AaCrz1* deletion resulted in antioxidant system disruption in *A. alternata*. Deletion of *ERG3* gene encoding the C-5 sterol desaturase has been reported to result in a failure of sterol synthesis, which alters the composition of cell membrane lipids and thus alters the anisotropy of the cell membrane to H_2_O_2_, allowing more H_2_O_2_ to enter the cell and leading to an increase in intracellular ROS levels [49,50,51,52]. RNA-seq results confirmed that *Erg3* (ncbi_29116487) was significantly down-regulated, so we hypothesized that this would lead to an increase in ROS levels in *ΔAaCrz1*. Intracellular accumulation of ROS caused oxidative damage to proteins, nucleic acids, and other cellular components [53,54], and these oxidative damages further acted as inducers of autophagy [55]. KEGG pathway enrichment analysis showed that DEGs were up-regulated in autophagy-other pathways, including several important proteins in the process of autophagy, such as ATG4 (ncbi_29109885), which plays a major role in macroautophagy [56], ATG9 (ncbi_29111274), an important source of membrane components required for early autophagosome formation [57], and ATG1 (ncbi_29117977), a key regulatory center protein kinase upstream of autophagy mechanism [58,59]. Taken together, we speculate that ROS accumulation in *ΔAaCrz1* induces an increase in autophagy level, although this warrants further exploration and examination.

Ca^2+^ is an essential regulator of many pathways in all eukaryotic cells, and disruption of calcium homeostasis would severely interfere with the related metabolic processes. Therefore, cells have evolved elaborate calcium homeostasis systems to regulate intracellular Ca^2+^ levels [60]. RNA-seq showed that two calcium transport genes, *pmc1* (ncbi_29117801, ncbi_29120094) and *VCX1* (ncbi_29118340, ncbi_29119593), located on the vacuolar membrane were significantly down-regulated in *ΔAaCrz1*. Studies have shown that *Pmc1* and *VCX1* knockout impaired intracellular calcium transport, resulting in a significant increase in cytosol calcium ion levels. In addition, *Pmc1* plays a key role in the pathogenesis of *Cryptococcosis* [61]. Calcineurin regulator (RCANs) (also known as calcium vasopressin) directly bind to the catalytic subunit of calcineurin and regulate the phosphatase activity of calcineurin [62]. In plant pathogenic fungi, RCAN deficiency severely reduced mycelial growth, spore formation and virulence [63,64,65]. From the RNA-seq results, we identified a structural gene belonging to this family, *RCAN2* (ncbi_29109809) (Appendix A), which was significantly down-regulated in *ΔAaCrz1*. Thus, intracellular calcium homeostasis in *A. alternata* was affected by *AaCrz1* knockdown, which is consistent with our previous results [10].

When fungal cells are exposed to various external stimuli, such as high temperature, low temperature, hyperosmolarity, oxidative stress, or antifungal drugs, Ca^2+^-mediated signaling induces an instantaneous increase in intracellular Ca^2+^ [66,67]. Free intracellular Ca^2+^ binds to calmodulin (CaM) to form a Ca^2+^/CaM complex to activate calcineurin (CaN), which further dephosphorylates the transcription factor *Crz1* to the nucleus [11,68]. *Crz1* containing the zinc finger domain can specifically bind to the CDREs sequence of about 24 bp to regulate the expression of target genes [16,69]. The STRING protein database was used to predict proteins that interact with AaCrz1. As shown in Figure 7, 30 proteins were screened.

Based on current investigation results, our previous findings in the phenotypic observation of *ΔAaCrz1*, and studies from other researchers, the mechanisms of *AaCrz1* regulating the biological function of *A. alternata* and the mode of *AaCrz1* knockout inducing changes in *A. alternata* were schematically illustrated (Figure 11). Intracellular free Ca^2+^ binds to CaM and CaN, and then the activated CaN dephosphorylates *Crz1* in the cytoplasm and enters the nucleus to regulate target gene expression [11,15,16]. The transcription factor *Crz1* is involved in several biological processes such as fungal vegetative growth, infection structure formation, cell wall integrity, calcium homeostasis, stress resistance, and virulence [10,17,20,21,34,35]. RNA-seq results showed that the antioxidant system of *A. alternata* was damaged and the ROS level was increased after *AaCrz1* knockout. In addition, the expression of *Erg3* was down-regulated, which promoted the entry of external H_2_O_2_ into the cell (Figure 8). The expression level of autophagy genes was up-regulated, which we hypothesized might be induced by elevated ROS levels in vivo. The expression levels of *atrA* and *atrB*, two genes responsible for drug efflux, were also decreased in *ΔAaCrz1*. In conclusion, the RNA-seq data provide clues for exploring the specific regulatory mechanism of *AaCrz1* in response to cuticular wax signaling involved in the infection structure differentiation and pathogenicity of *A. alternata*.

## 4. Material and Methods

### 4.1. Fungal Strains

The wild type (WT) and *ΔAaCrz1* strains of *A. alternata* used in this study have been previously characterized [10].

### 4.2. Wax Extraction from “Zaosu” Pear Peel

According to the method of Tang et al. [3], “Zaosu” pear specimens without the presence of pests and diseases were selected, cleaned, and dried at room temperature. The pear fruit was completely immersed in chloroform solution for 30 s, and then the chloroform enriched with wax extract was placed in a constant temperature vacuum distillation device to remove chloroform and obtain wax crude extract. The wax crude extract was blown to powder using a nitrogen blowing apparatus and stored at 4 °C for subsequent experiments.

### 4.3. Observation of the Infection Structure Formation of A. alternata Induced by Pear Wax in Onion Epidermis

According to the method of Yang et al. [10], the onion was cut into a size of 2 cm × 2 cm and the epidermal membrane was removed. The onion membrane was placed in chloroform for 1 min for dewaxing, and then the fruit wax extract was evenly spread with a spreader. Then, 20 μL spore suspensions of the WT, *ΔAaCrz1*, and *ΔAaCrz1-C* specimens were added to the treated onion membrane, respectively. The process of *A. alternata* infection structure formation was observed under an optical microscope, and the spore germination rate (%), appressorium formation rate (%), and infection hyphae formation rate (%) were calculated at 2 h, 4 h, 6 h, and 8 h of incubation at 28 °C, respectively. Each treatment had three parallels, and 100 conidia were counted each time.

### 4.4. Sample Preparation, RNA Extraction, Library Construction and Sequencing for Transcriptomes

Spore suspensions of the WT and *ΔAaCrz1* strains at a concentration of 1 × 10^6^ conidia mL^−1^ were prepared and then inoculated on the hydrophobic surface of the Gelbond PAG membrane coated with pear wax extract. The hydrophobic membrane containing the spore suspension was carefully placed in a petri dish with a moist circular filter paper (used to provide the humidity required for spore germination) at the bottom and placed in a dark incubator at 28 °C. After 6 h, the spores on the Gelbond PAG membrane were rapidly collected into a centrifuge tube to collect spore precipitation, then frozen in liquid nitrogen and stored at −80 °C for RNA-seq. Each treatment was repeated three times. The six samples of WT and *ΔAaCrz1* were named WT-1, WT-2, WT-3, M-1, M-2, and M-3, respectively. Total RNA was extracted using the Trizol Kit (Invitrogen, Carlsbad, CA, USA) with some modifications according to the manufacturers’ instructions. The quality of extracted RNA was examined using agarose gel electrophoresis, a NanoPhotometer spectrophotometer, a Qubit 2.0 Fluorometer, and an Agilent 2100 bioanalyzer to fulfill the database construction requirements. RNA libraries were sequenced on the Illumina sequencing platform by Genedenovo Biotechnology Co., Ltd. (Guangzhou, China).

### 4.5. Reference Genomes and Reference Genes

Since a reference genome was already available for *A. alternata* [52], the clean data for the samples WT-1, WT-2, WT-3, M-1, M-2, and M-3 were compared with the reference genome and reference genes, respectively.

### 4.6. Functional Annotation and GO and KEGG Classification

False Discovery Rate (FDR) ≤ 0.05 and FoldChange ≥ 2 were used as screening criteria for differential expression analysis. Databases used for transcriptome analysis included GO (https://www.geneontology.org/, accessed on 16 October 2023) and KEGG (https://www.genome.jp/kegg, accessed on 16 October 2023).

### 4.7. Interaction Network Analysis

The STRING protein interactions database (http://string-db.org, accessed on 25 October 2023) was applied to analyze the protein interactions of differential genes. For species included in the database, the set of differential genes was extracted from the database and cytoscape was used to construct the interactions network map; for species not included in the database, the sequences in the target gene set were firstly exposed to blast comparison with the protein sequences of the reference species included in the STRING database, and the protein interactions of this reference species determined by comparison were used to construct the interactions network.

### 4.8. Gene Expression Validation by RT-qPCR

The target gene sequences of *A. alternata* were obtained from NCBI. The primer sequences are shown in Appendix A. Quantitative RT-qPCR experiments were performed using the Takara Bioengineering (Dalian, China) kit. The PrimeScript™ RT reagent Kit with gDNAEraser (Perfect Real Time) cDNA synthesis kit (RR047A) was used to obtain cDNA. Fluorescence quantification was performed using the abm^®^Eva Green qPCR MasterMix-ROX kit. The PCR reaction system and PCR conditions are listed in Appendix A. The relative expression levels of the genes were calculated using the 2^−ΔΔCt^ method.

### 4.9. ROS Staining Observation

ROS staining of spores was performed using 2,7-dichlorodi-hydro fluorescein diacetate (DCFH-DA, D6470, Solarbio Science & Technology Co., Ltd., Beijing, China). DCFH-DA staining solution with a concentration of 1 g L^−1^ was prepared with DMSO reagent and stored at −20 °C for later use. A total of 1 mL spore suspension (1 × 10^6^ spores mL^−1^) of the WT and *ΔAaCrz1* mutant were placed in a 1.5 mL centrifuge tube and centrifuged at 8000× *g* and 4 °C for 10 min, and the supernatant was collected. The precipitate was suspended in 1 mL of PBS (0.01 M, pH 7.2–7.4) buffer, and 10 μL of DCFH-DA staining solution was added, and stained for 30 min in a 37 °C incubator protected from light. After two washes with 1 mL of PBS (0.01 M, pH 7.2–7.4) buffer and resuspension, 20 μL was aspirated and placed under a fluorescence microscope (CX21FS1C, Olympus, Tokyo, Japan) and photographed. The fluorescence intensity of spores was measured using ImageJ 2.

### 4.10. Determination of H_2_O_2_ Content

H_2_O_2_ content was determined using kits (Beijing Solarbio Science & Technology Co., Ltd., BC3595, Beijing, China). The spores were broken by ultrasonic wave (power 20%, ultrasonic 3 s, interval 10 s, repeated 30 times), followed by centrifugation at 8000× *g* and 4 °C for 10 min, and the supernatant was used to determine the content of H_2_O_2_. H_2_O_2_ content was determined according to the kit instructions, at 415 nm absorbance and expressed as μmol/10^4^ cell.

### 4.11. Statistical Analysis

Data for transcriptome analysis were plotted using the OmicShare tool (https://www.omicsshare.com/tools, accessed on 8 October 2023). The mean and standard deviation of RT-qPCR data were calculated using Microsoft Office 2010, and the data were plotted using Origin 8.5. Duncan’s multiple difference analysis was performed using SPSS 26.0 (*p* < 0.05).

## 5. Conclusions

In summary, *AaCrz1* contributes to the infection structure formation in *A. alternata* during the early infection stage. Moreover, our results provide clues for exploring the specific regulatory mechanism of *AaCrz1* in response to cuticular wax signaling involved in the infection structure differentiation and pathogenicity of *A. alternata*, especially involving a series of metabolic pathways such as cell wall degrading enzymes, ABC transporters, antioxidant capacity, autophagy, and ion homeostasis. The RNA-seq results will provide data support for searching for potential genes or proteins in *A. alternata* as targets for black spot disease control. Moreover, the findings will contribute to the development of target-based fungicides for postharvest disease control using molecular techniques such as synthetic dsRNA technology.

## Figures and Tables

**Figure 1 ijms-25-11950-f001:**
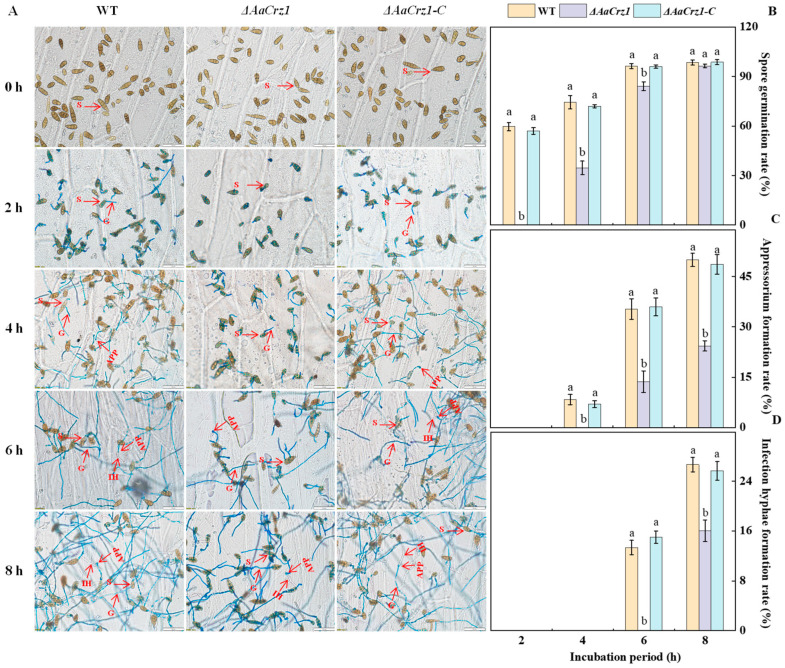
Effect of *AaCrz1* on the infection structure differentiation of *A. alternata* on the onion epidermis coated with fruit wax extract. Morphological observation of infection structure of the WT, *ΔAaCrz1*, and *ΔAaCrz1-C* strains on onion epidermis coated with fruit wax extract. S represents spores, G represents germ tube, APP represents appressorium, and IH represents infection hyphae (**A**); the spore germination rate of the WT, *ΔAaCrz1* and *ΔAaCrz1-C* strains (**B**); the spore appressorium formation rate (**C**) of the WT, *ΔAaCrz1*, and *ΔAaCrz1-C* strains; the infection hyphae formation rate of the WT, *ΔAaCrz1*, and *ΔAaCrz1-C* strains (**D**). Each treatment was in parallel and repeated three times. Vertical lines indicate the standard error of the means. Different letters indicate significant differences (*p* < 0.05).

**Figure 2 ijms-25-11950-f002:**
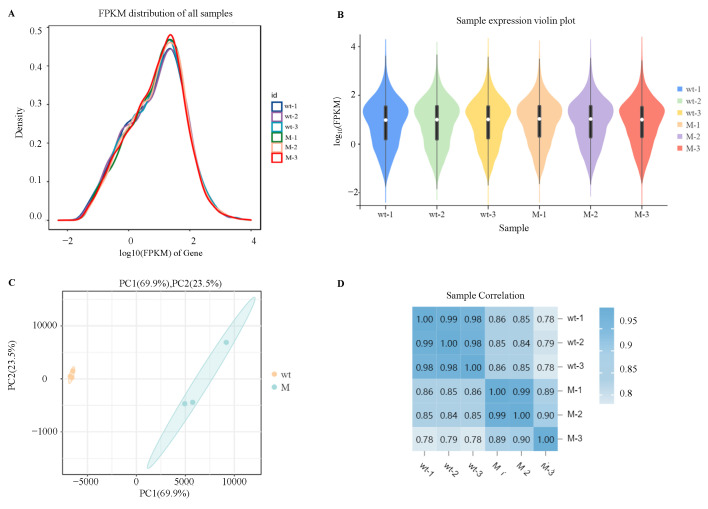
Analysis of gene expression abundance distribution and sample relationship. (**A**) Abundance distribution map of gene expression in each sample; (**B**) Violin plot of gene expression in each sample. The white dots represent the median, and the black rectangle indicates the interquartile range. The length of thin black lines indicates the degree of dispersion and symmetry of the non-abnormal data, and the external shape is the kernel density estimate. The vertical axis length of the plot shows the degree of data dispersion, and the horizontal axis length reflects the amount of data distribution at a certain vertical axis position; (**C**) Sample principal component analysis (PCA). The PC1 coordinate represents the first principal component, and the percentage in parentheses represents the contribution of the first principal component to the sample difference; The PC2 coordinate represents the second principal component, and the percentage in parentheses represents the contribution of the second principal component to the sample difference; (**D**) Sample correlation analysis. The number in the lattice is the Pearson correlation coefficient. The bluer the color is, the greater the correlation coefficient is, and the stronger the sample similarity is. Note: wt represents the WT strain and M represents the *ΔAaCrz1* strain.

**Figure 3 ijms-25-11950-f003:**
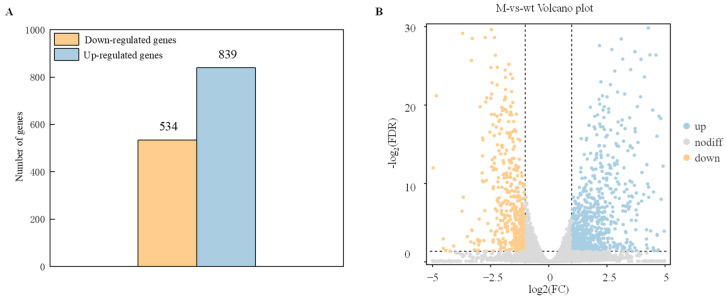
Statistics of differential expressed genes (DEGs). (**A**) The statistical map of DEGs. The yellow column represents the down-regulated differential expressed genes (DEGs), and the blue column represents up-regulated differentially expressed genes (DEGs). (**B**) The volcano plot of DEGs. The abscissa represents the logarithm of the difference between the two groups, and the ordinate represents the −log_2_ value of the FDR of the difference between the two groups. Blue (up-regulated expression) and yellow (down-regulated expression) points indicate differential gene expression, and gray points indicate no difference.

**Figure 4 ijms-25-11950-f004:**
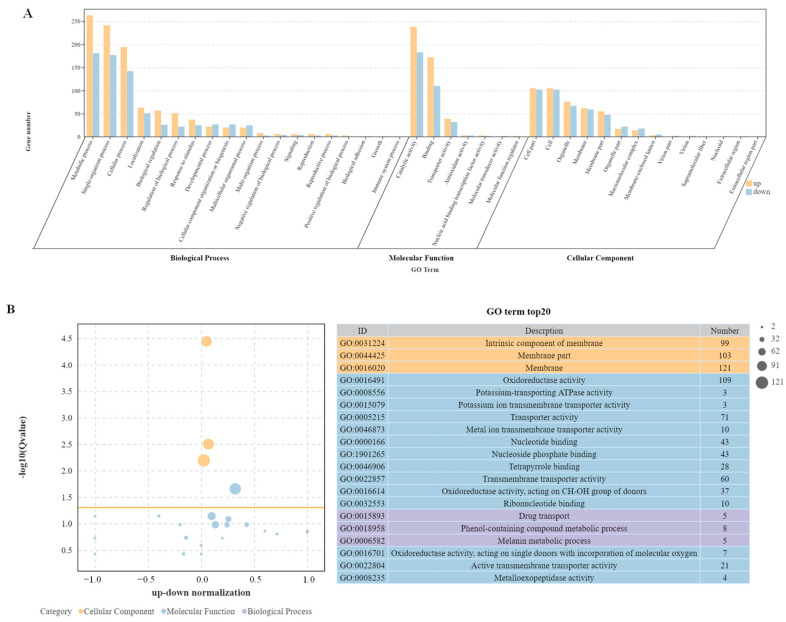
Enrichment analysis of GO terms. (**A**) The Go enrichment of DEGs in the WT-vs-*ΔAaCrz1* mutant strain group (the yellow column represents the up-regulated genes, the blue column represents the down-regulated ones). (**B**) Top 20 most significantly enriched GO terms by −log_10_(*Q*-value).

**Figure 5 ijms-25-11950-f005:**
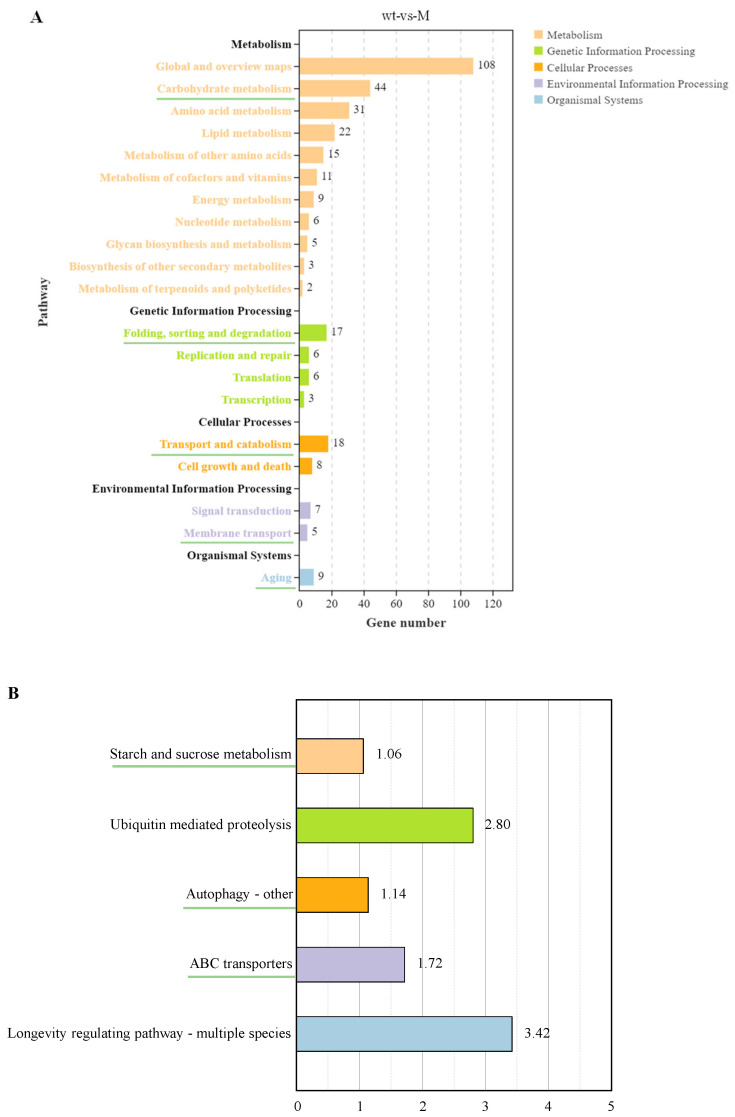
Enrichment analysis of the KEGG pathway. (**A**) The enrichment of KEGG pathways. (**B**) Five pathways selected from five secondary annotations underlined in Figure 5A by −log_10_ (*Q*-value).

**Figure 6 ijms-25-11950-f006:**
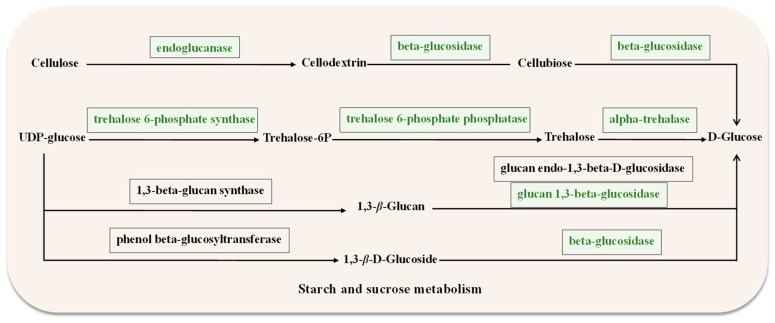
Starch and sucrose metabolism diagram. The box represents the enzymes required for the metabolism, and the green box indicates the downregulated genes encoding the corresponding enzymes.

**Figure 7 ijms-25-11950-f007:**
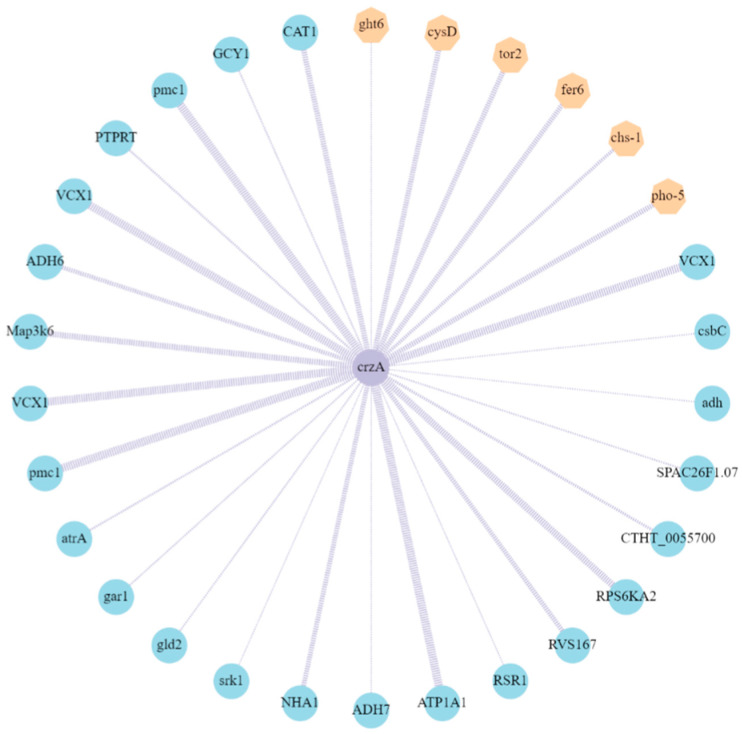
Identification of proteins interacting with *AaCrz1*. The yellow hexagons represent the proteins encoded by upregulated genes, whereas green circles represent the proteins encoded by downregulated genes in *ΔAaCrz1* compared to WT. The thickness of the purple line indicates the level of the score. The greater the score, the higher the interaction probability.

**Figure 8 ijms-25-11950-f008:**
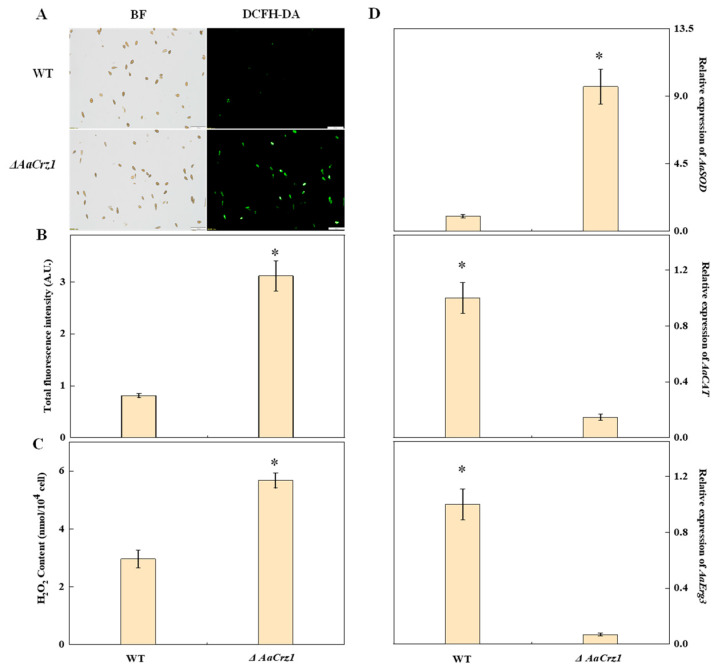
Effects of *AaCrz1* deletion on ROS level in *A. alternata*. (**A**) ROS staining of spores. BF: Bright field; DCFH-DA: 2,7-dichlorodi-hydrofluorescein diacetate. (**B**) The fluorescence intensity of spores. (**C**) H_2_O_2_ content. (**D**) The expression of ROS-related genes. Bars indicate standard errors. Each treatment was performed in parallel and repeated three times. * indicates significant differences (* *p* < 0.05).

**Figure 9 ijms-25-11950-f009:**
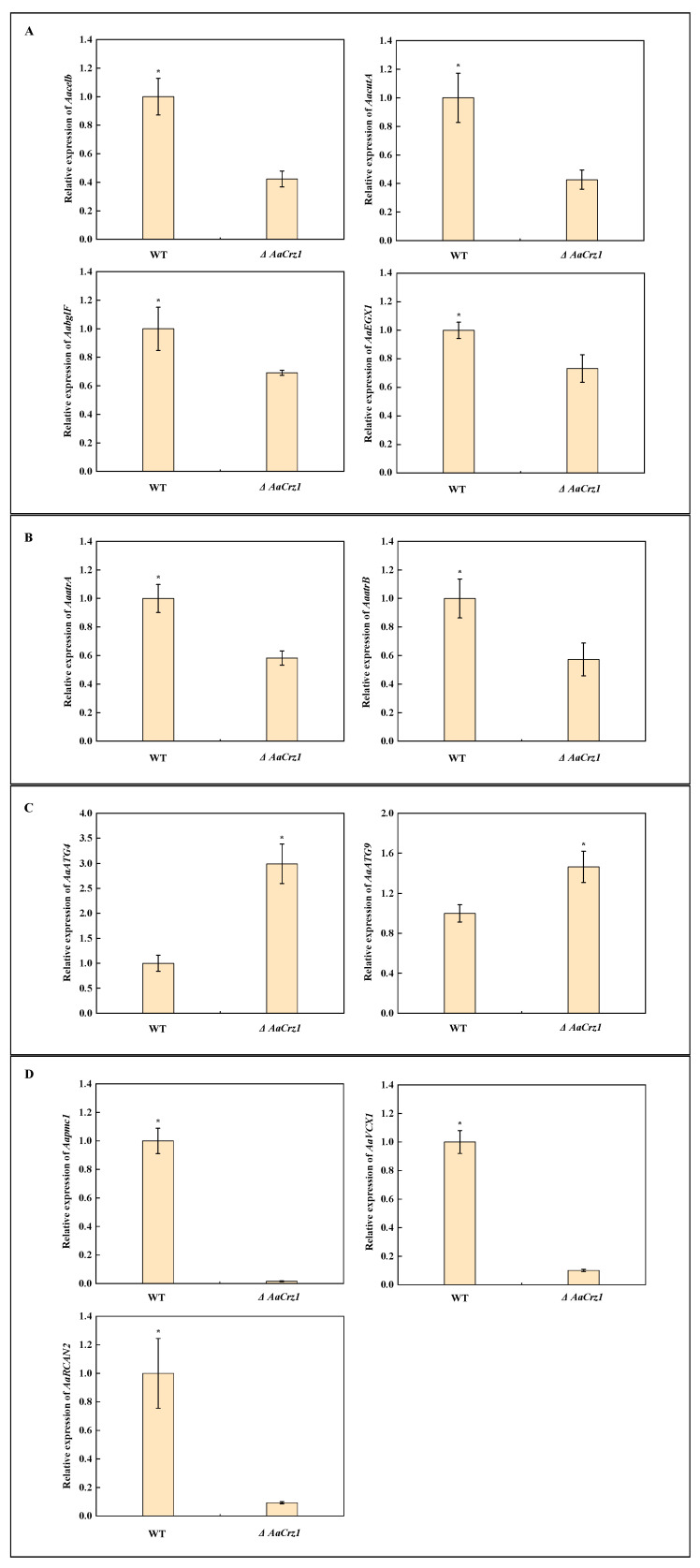
The relative expression level change of 11 selected genes. (**A**) Genes from cell wall degrading enzymes. (**B**) Genes from ABC transporter. (**C**) Genes from autophagy. (**D**) Genes from calcium homeostasis. Asterisks denote significant differences (* *p* < 0.05).

**Figure 10 ijms-25-11950-f010:**
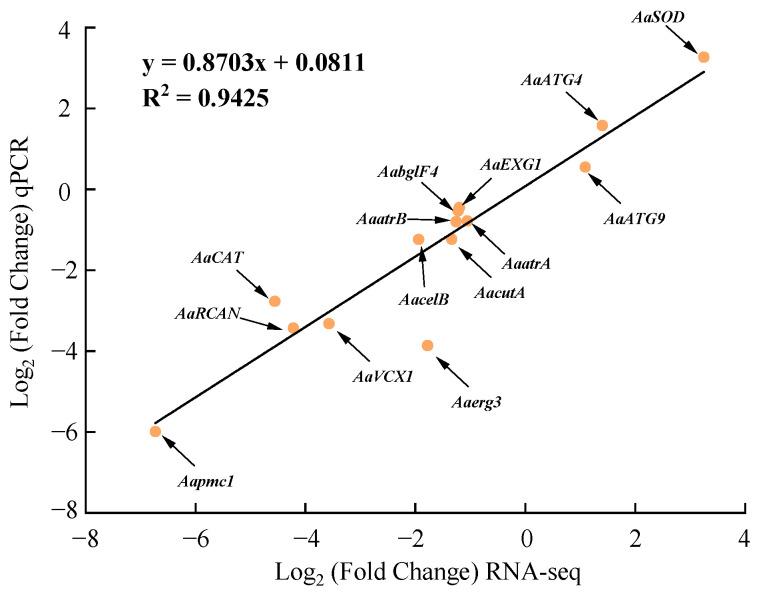
Regression analysis of the expression of 14 DEGs detected by RNA-seq and RT-qPCR.

**Figure 11 ijms-25-11950-f011:**
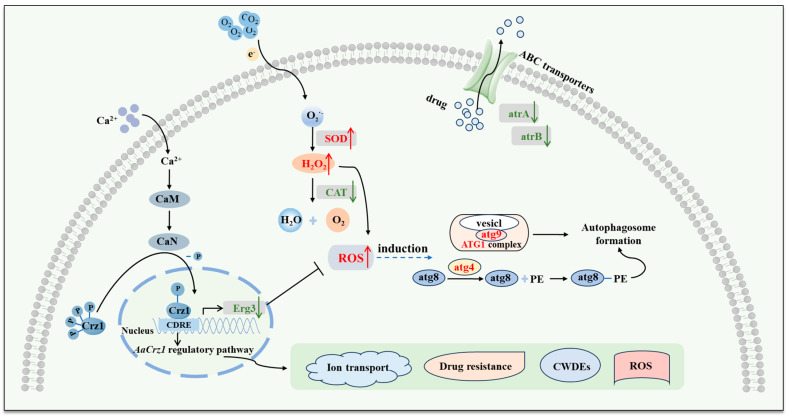
The mechanism of *AaCrz1* regulating the biological functions of *A. alternata,* and the pattern of in vivo changes in *A. alternata* induced by *AaCrz1* knockout. The red font and red arrows represent up-regulated gene expression or increased substance content, and the green font and green arrows represent down-regulated gene expression in the figure.

## Data Availability

Data are contained within the article and Appendix A.

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
