# Peer review of "Transcriptome Analysis Provides Insights into the Mechanism of the Transcription Factor AaCrz1 Regulating the Infection Structure Formation of Alternaria alternata Induced by Pear Peel Wax Signal"

_ijms, 2024, doi:10.3390/ijms252211950_

Round 1

Reviewer 1 Report

Comments and Suggestions for Authors

The manuscript presented could have important applications in protecting against post-harvest pathogen attacks and is, undoubtedly, of interest.

However, in my opinion, the work is still incomplete and not entirely convincing.

I would suggest the following improvements:

  • The description of Crz1 is insufficient. It should be expanded to include more references regarding the function and discovery of this transcription factor.

  • The figures need to be consolidated, as Figure 1 is poorly arranged, making the manuscript difficult to read. The figures should be well-centered and clearly presented.

  • There are several formatting errors, such as those found in lines 12, 30, 35, etc., that need to be revised.

  • The relevance of this study is not fully explained.

  • Why do the authors use FPKM for RNA-seq normalization when it is generally accepted that this method does not account for mRNA lengths, which could impact the expression analysis?

  • The conclusion drawn from Figure 7 is not supported by an experiment demonstrating ROS accumulation. For example, there are commercial kits available that facilitate easy visualization of this effect under a microscope, or by measuring fluorescence/absorbance using spectrometry. Additionally, alterations in calcium homeostasis should also be tested.

  • The data presented are used to predict conclusions and models that are primarily obtained in silico, and some of these predictions seem speculative. The proposed mechanisms for infection should be demonstrated molecularly.

In conclusion, I believe the research presented requires significant improvement and, in its current state, is not suitable for publication.

Comments on the Quality of English Language

minor errors

Reviewer 2 Report

Comments and Suggestions for Authors

In this manuscript, the authors investigated the gene expression pattern in wild-type and AaCrz1 null mutant Alternaria alternata and speculated on the possible role of AaCrz1 in wax signal-induced response. The results help understand the role of AaCrz1 in Alternaria alternata. Nevertheless, the data are still very preliminary, so they cannot build a complete story.

 1.     An important control was absent from the experiment. The authors detected infection structure formation in the onion epidermis coated with pear peel wax. Data on infection structure formation and gene expression in onion epidermis that was not coated with pear peel wax should be supplemented.

2.     The introduction section is too short. Related research progress was not adequately shown.

3.     The images in Figure 1 should be reorganized. Image B, C, and D should be placed more closely together.

4.     In Figure 2, words beside the axis were too small to be recognized. The size and font of the words in different images were not identical. In Figure 2 C, what do PC1 and PC2 mean? It should be described in the figure legend.

5.     In Figure 3, the up-regulated and down-regulated genes should be marked in the same color in different images.

6.     Lines 134, “1006 up-regulated DEGs and 723 down-regulated DEGs” is not understandable. Based on data in Figure 3, only 893 DEGs were up-regulated, and 534 DEGs were down-regulated.

7.     Figure 7: I dont agree that such a schematic image is appropriate since most pathways in the figure are speculative. More solid data should be provided before drawing a signaling network like this.

8.     The binding site of AaCrz1 in target genes’ promoters should be analyzed or at least discussed. It should be addressed among the up- or down-regulated DEGs, which are regulated directly by AaCrz1, which are regulated indirectly.

9.     Lines 305-315 contain repetitive statements of the result, and they should be removed.

10.     Line 318, “CWED” should be corrected to “CWDE.

Round 2

Reviewer 2 Report

Comments and Suggestions for Authors

The revised manuscript lookes better than the original version.  The authors response to the comments properly. 

Author Response

Thank you for your valuable evaluation of our work.